# TOWARDS COUNTERFACTUAL FAIRNESS THROUGH AUXILIARY VARIABLES

**Bowei Tian**[1], **Ziyao Wang**[1], **Shwai He**[1], **Wanghao Ye**[1]
**Guoheng Sun**[1], **Yucong Dai**[2], **Yongkai Wu**[2*], **Ang Li**[1*]
[1]University of Maryland, College Park, [2]Clemson University
{btian1, ziyaow, shwaihe, wy891, ghsun, angliece}@umd.edu
yucongd@g.clemson.edu, yongkaw@clemson.edu

## ABSTRACT

The challenge of balancing fairness and predictive accuracy in machine learning models, especially when sensitive attributes such as race, gender, or age are considered, has motivated substantial research in recent years. Counterfactual fairness ensures that predictions remain consistent across counterfactual variations of sensitive attributes, which is a crucial concept in addressing societal biases. However, existing counterfactual fairness approaches usually overlook intrinsic information about sensitive features, limiting their ability to achieve fairness while simultaneously maintaining performance. To tackle this challenge, we introduce **EXO**genous **C**ausal reasoning (EXOC), a novel causal reasoning framework motivated by exogenous variables. It leverages auxiliary variables to uncover intrinsic properties that give rise to sensitive attributes. Our framework explicitly defines an auxiliary node and a control node that contribute to counterfactual fairness and control the information flow within the model. Our evaluation, conducted on synthetic and real-world datasets, validates EXOC's superiority, showing that it outperforms state-of-the-art approaches in achieving counterfactual fairness. Our code is available at https://github.com/CASE-Lab-UMD/counterfactual_fairness_2025.

## 1 INTRODUCTION

Machine learning has been widely adopted in prediction tasks (Brennan et al., 2009; Corbett-Davies et al., 2023) such as personalized recommendation (Mehrotra et al., 2018; Wu et al., 2021) and image classification (Bhojanapalli et al., 2021; Chen et al., 2021). Recent literature shows that predictions based on traditional machine learning methods often exhibit bias against certain demographic subgroups, which are described by sensitive attributes such as race, gender, age, and sexual orientation. Therefore, developing a fairer predictor has attracted considerable attention (Bellamy et al., 2019; Bird et al., 2019; Caton & Haas, 2024). Among them, *counterfactual fairness* applies causal mechanisms to model how discrimination occurs and measure societal bias at an individual level, using Pearl's causal structural models (Pearl, 2009). The idea behind counterfactual fairness is to ensure that predictions from the same individual remain consistent even if their sensitive attribute would have changed. Kusner et al. (2017) introduce the framework for counterfactual fairness at the individual level using causal models. After that, several works focus on counterfactual fairness. Russell et al. (2017) propose a new counterfactual fairness framework by integrating multiple counterfactual assumptions, aiming to address inconsistencies in fairness across different causal models. The paper uses a Bayesian approach to unify different counterfactual assumptions into a probabilistic model, thereby better handling complex fairness issues. Wu et al. (2019) presents a unified definition that covers most of the previous causality-based fairness notions, namely the path-specific counterfactual fairness (PC fairness), and proposes an estimation approach for unidentified causal quantities. Ma et al. (2023) propose a method to achieve counterfactual fairness without requiring a predefined causal graph by learning directly from observational data. The approach involves creating a counterfactually fair dataset through augmentation and using a carefully designed loss function to ensure fairness during model training.

We observe that the majority of existing methods for counterfactual fairness focus on analyzing causal inference and their counterfactual framework (Kusner et al., 2017; Russell et al., 2017) or creating a counterfactual-fair augmentation dataset that is agnostic to a casual graph (Ma et al., 2023), which can be inferred directly using the augmentation dataset and crafted loss design. However, to the best of our knowledge, most existing works assume sensitive attributes should not be causally influenced by any other variables (Kusner et al., 2017; Berk et al., 2021; Ma et al., 2023). This assumption overlooks the essentials of sensitive features, i.e., which part of the sensitive feature is intrinsic or essential for the inference and which part should be neglected. Also, existing methods usually fit into a specific scenario where the causal relationship from the sensitive attribute to the target attribute is fixed. For example, race should generally not influence decision-making at all, making it hard to extend and distribute in real-world scenarios. For example, in a demography experiment, race distribution can be deduced from the geographic distribution of the population, which can not be causally neglected.

To tackle these challenges, we propose a novel framework, EXOC, which introduces intuitive modifications to the causal model. This framework utilizes the auxiliary variables in causal inference, extracting essential information from sensitive attributes and effectively controlling the flow of information from the sensitive attribute to the target attribute. We summarize our contributions as follows:

- We develop a framework that utilizes the auxiliary variables in causal inference, extracting essential information from sensitive attributes and enhancing fairness without sacrificing much accuracy.
- We formalize a method to regulate the flow of information from the sensitive attribute to the target attribute, effectively controlling the balance between accuracy and fairness.
- We provide theoretical analysis and conduct extensive baseline and ablation experiments to validate the effectiveness of our approach.

## 2 PRELIMINARIES

### 2.1 COUNTERFACTUAL FAIRNESS

Counterfactual fairness (Kusner et al., 2017) is an individual-level fairness notion based on the causal model. It is constructed on the Pearl's causal framework (Pearl, 2009), which is defined as a triple $(U, V, F)$ so that:

- $U$ is a set of latent background variables, which are exogenous and not caused by any variable in the set $V$;
- $V$ is a set of observed variables, which are endogenous and determined by $U \cup V$;
- $F$ is a set of functions $\{f_1, ..., f_n\}$, on for each $V_i \in V$, so that $V_i = f_i(pa_i, U_{pa_i})$, where $pa_i \subseteq V \setminus \{V_i\}$ and $U_{pa_i} \subseteq U$ are variables that directly determine $V_i$.

A causal model is associated with a causal graph, which is a directed acyclic graph (DAG). Each node in the causal graph represents a variable in the causal model, and each directed edge corresponds to a causal relationship. In the causal model, the counterfactual estimands are facilitated by interventions through *do*-calculus, which simulates the physical interventions that force some variables to take certain values. For example, for observed variables $A$ and $B$, the value of the counterfactual "what would $A$ have been if $B$ had been $b$" is denoted by $A_{B \leftarrow b}$.

**Counterfactual fairness (Kusner et al., 2017; Wu et al., 2019):** Given a factual condition $\mathbf{O} = \mathbf{o}$, the predictor $Y = f(\mathbf{O})$ is *counterfactually fair* if under any context $\mathbf{o}$,

$$P(Y_{S \leftarrow s} = y \mid \mathbf{o}) = P(Y_{S \leftarrow s'} = y \mid \mathbf{o}), \quad \forall s' \neq s, \tag{1}$$

where $\mathbf{O} = \{S, \mathbf{X}\}$, $S$ is the sensitive attribute and $\mathbf{X}$ is observed non-sensitive attributes.

**Approximate counterfactual fairness (Russell et al., 2017):** A predictor $Y = f(\mathbf{O})$ satisfies $(\delta, 0)$-*approximate counterfactual fairness* if, given the factual condition $\mathbf{O} = \mathbf{o}$, we have:

$$|[(Y_{S \leftarrow s} - Y_{S \leftarrow s'}) \mid \mathbf{o}]| \leq \delta, \quad \forall s' \neq s. \tag{2}$$

This approximate metric measures counterfactual fairness in practical manners. Unless otherwise specified, we refer to this approximate metric as counterfactual fairness in our theoretical analysis.

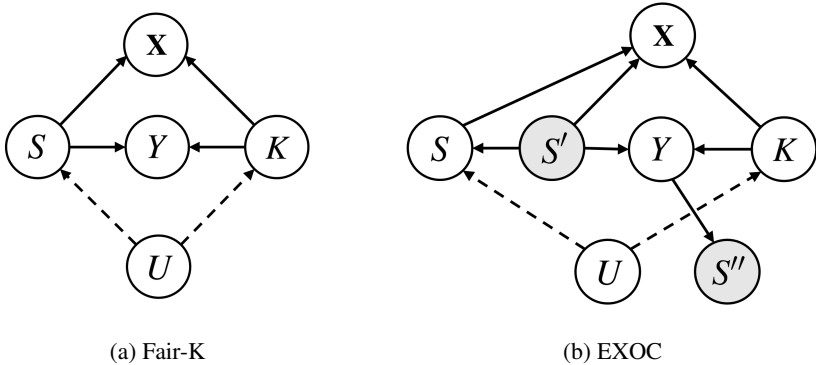

(a) Fair-K                                    (b) EXOC

Figure 1: The causal models of Fair-K and EXOC. $S$ is the sensitive attribute, $\mathbf{X}$ is observed non-sensitive attributes, $Y$ is the target attribute, $K$ is the latent domain knowledge, and $S'$ and $S''$ are latent auxiliary nodes, $U$ is the exogenous variable. The solid lines represent designed causal relationships, and dashed lines mean our focused existing relationships in implementation, illustrated in 3.2.2 (note that $U$ have existing causal relationships with every node (Pearl, 2009)).

## 2.2 COUNTERFACTUALLY FAIR LEARNING APPROACHES

In the counterfactually fair machine learning literature, Fair-K (Kusner et al., 2017; Ma et al., 2023) is a widely adopted framework. As illustrated in Fig. 1a, it is designed on the Law school dataset (Krueger et al., 2021) and assumes a node $K$ representing domain knowledge that can act as non-deterministic causes of $\mathbf{X}$. Then, it trains a predictor using $K$ to predict $\hat{Y}$, e.g., using logistic regression and achieves a counterfactual fairness improvement. However, the causal effect transmitting from exogenous variables $U$ is not fully utilized in the deployments of counterfactually fair predictors, potentially leading to significant performance decreases.

## 3 DESIGN

### 3.1 CAUSAL MODEL OVERVIEW

To address the above limitations, we propose EXOC, a novel framework that introduces the auxiliary node and the control node, which instantiates the exogenous variable $U$. Specifically, the model leverages controllable auxiliary nodes $S'$ to simultaneously capture intrinsic, latent information from $\mathbf{X}$, $Y$, and $S$. The model enhances the overall performance and fairness balance by incorporating this additional auxiliary compared to Fair-K in Fig. 1a. Motivated by the controllable nature of $S'$, we further introduce $S''$, with the aim that $S''$ can support $S'$ in controlling the balance between fairness and accuracy. Strengthening the relationship between $S'$ and $S''$ indicates a stronger alignment with fairness, whereas weakening allows $S'$ to tap into more intrinsic information from $S$ and emphasize performance. Through this flexible mechanism, we can prioritize fairness or accuracy in various real-world scenarios. We explain the components of the EXOC framework in the following subsections.

### 3.2 $S'$: THE AUXILIARY NODE

#### 3.2.1 ILLUSTRATING AUXILIARY NODE $S'$ IN A SIMPLIFIED CASE

We observe that previous works fail to dive into the essential of sensitive features and consider $U$ as unknown background variables in a causal model. Rethinking the role of exogenous variable $U$, we devise an idea to instantiate $U$ into an auxiliary node $S'$ in the model. To demonstrate how the auxiliary node $S'$ achieves counterfactual fairness, we first take a simple example: consider the ideal linear model (the model can ideally fit the determination of the real world, where $U$ is eliminated) with normal distributions. In Fig. 1a, for each *individual*, the causal relationship to $Y$ can be written

as:

$$Y = \alpha S + \beta K, \tag{3}$$

where $\alpha$ and $\beta$ are path coefficients (Pearl, 2009). Therefore, in causal inference,

$$Y_{S \leftarrow s} \mid \mathbf{o} = \alpha \cdot (S_{S \leftarrow s} \mid \mathbf{o}) + \beta \cdot (K \mid \mathbf{o}) = \alpha s + \beta k, \quad k \sim \mathcal{N}(\mu_K, \sigma_K^2), \tag{4}$$

$$(Y_{S \leftarrow s^*} - Y_{S \leftarrow s} \mid \mathbf{o})_a = \alpha(s^* - s) + \beta(k_1 - k_0), \tag{5}$$

where the same alphabet with different subscript numbers is sampled from the same corresponding distribution, $\alpha(s^* - s)$ is fixed, and $\beta(k_1 - k_0) \sim \mathcal{N}(0, 2\beta^2 \sigma_K^2)$.

Similarly in Fig. 1b, we have:

$$Y = \tilde{\alpha} S' + \tilde{\beta} K, \tag{6}$$

where the alphabet with the tilde operator has similar meanings. Since the model structure differs between Fig. 1a and Fig. 1b, we distinguish their values using tilde. Therefore,

$$Y_{S \leftarrow s} \mid \mathbf{o} = \tilde{\alpha} \cdot (S'_{S \leftarrow s} \mid \mathbf{o}) + \tilde{\beta} \cdot (K \mid \mathbf{o}) = \tilde{\alpha} s' + \tilde{\beta} \tilde{k}, \quad s' \sim \mathcal{N}(\tilde{\mu}_{S'}, \tilde{\sigma}_{S'}^2), \quad \tilde{k} \sim \mathcal{N}(\tilde{\mu}_K, \tilde{\sigma}_K^2), \tag{7}$$

$$(Y_{S \leftarrow s^*} - Y_{S \leftarrow s} \mid \mathbf{o})_b = \tilde{\alpha}(s'_1 - s'_0) + \tilde{\beta}(\tilde{k}_1 - \tilde{k}_0), \tag{8}$$

apply Three Sigma Rule (Pukelsheim, 1994) on these results:

$$(Y_{S \leftarrow s^*} - Y_{S \leftarrow s} \mid \mathbf{o})_a^{\pm 3\sigma} = (\alpha(s^* - s) + \beta(k_1 - k_0))^{\pm 3\sigma} \tag{9}$$

$$= \alpha(s^* - s) \pm 3\sqrt{2} \cdot |\beta| \sigma_K, \tag{10}$$

$$(Y_{S \leftarrow s^*} - Y_{S \leftarrow s} \mid \mathbf{o})_b^{\pm 3\sigma} = (\tilde{\alpha}(s'_1 - s'_0) + \tilde{\beta}(\tilde{k}_1 - \tilde{k}_0))^{\pm 3\sigma} \tag{11}$$

$$= \pm 3 \sqrt{2 \left( \tilde{\alpha}^2 \tilde{\sigma}_{S'}^2 + \tilde{\beta}^2 \tilde{\sigma}_K^2 \right)}, \tag{12}$$

where these bounds are not exceeded in 99.7% of cases, so the exceptions are negligible. Therefore, we can estimate the upper bound of $|[Y_{S \leftarrow s^*} - Y_{S \leftarrow s} \mid \mathbf{o}]|$, i.e., approximate counterfactual fairness bound in these equations as:

$$|[Y_{S \leftarrow s^*} - Y_{S \leftarrow s} \mid \mathbf{o}]_a| \leq \left| \alpha(s^* - s) \pm 3\sqrt{2} \cdot |\beta| \sigma_K \right|$$

$$= |\alpha(s^* - s)| + 3\sqrt{2} \cdot |\beta| \sigma_K = \delta_a, \tag{13}$$

$$|[Y_{S \leftarrow s^*} - Y_{S \leftarrow s} \mid \mathbf{o}]_b| \leq 3 \sqrt{2 \left( \tilde{\alpha}^2 \tilde{\sigma}_{S'}^2 + \tilde{\beta}^2 \tilde{\sigma}_K^2 \right)} = \delta_b, \tag{14}$$

where the value corresponds to $\delta$ in the approximate counterfactual fairness definition. As $\alpha(s^* - s)$ is the counterfactual parity that plays a more important role than the standard deviation, we showcase that $\delta_a > \delta_b$, so the scenario in EXOC is tighter than Fair-K in the constraint of counterfactual fairness. Therefore, $S'$ theoretically helps improve the counterfactual fairness in this case.

### 3.2.2 EXTENDING AUXILIARY NODE $S'$ INTO A GENERAL CASE

To mitigate the strong assumptions that the model is ideal and the distribution of nodes is normal, we use arbitrary functions for the causal model. We have $Y = f(S, K, U)$ in Fair-K and $Y = f(S', K, U)$ in EXOC. When we calculate the counterfactual fairness, we will similarly operate a counterfactual parity between different sensitive attributes as in Eq. 5 and 8, i.e., $(f(S, K, U)_{S \leftarrow s} - f(S, K, U)_{S \leftarrow s^*}) \mid \mathbf{o}$ in Fair-K and $(f(S', K, U)_{S \leftarrow s} - f(S', K, U)_{S \leftarrow s^*}) \mid \mathbf{o}$ in EXOC. As $S'$ is a non-descendant of $S$, there should also be an elimination of $S$ parity when calculating the counterfactual fairness in EXOC. Therefore, we expect a promotion of counterfactual fairness.

Since the analysis based on counterfactuals from Pearl's SCM framework (Pearl, 2009) conflates the predictor $\hat{Y}$ with the outcome $Y$ (Kusner et al., 2017), and it does not explicitly incorporate probabilistic deployment of causal models. So, when we reconsider causality from the information perspective, we discover that $S'$ not only achieves counterfactual fairness but also has the nature of controlling information flows from $S$ to $Y$ and from $K$ to $Y$.

Specifically, in the deployment of EXOC, we first perform inference on the model using an observed training set to estimate the posterior distribution of $P(K \mid \mathbf{O})$. Subsequently, we train the logistic regression predictor $\hat{Y} = \varsigma(K)$ to model the relationship between $\mathbf{O}$ and $Y$. This predictor $\varsigma(\cdot)$ focuses on the correlation without the craft of causality and this ignorance of causality can be depicted as the impact from the exogenous variable $U$, focusing on two causal relations of $U \to K$ and $U \to S$. They yield a backdoor path $\pi_\vartheta = \{K \leftarrow U \to S\}$. Without loss of generality, we conduct a path analysis of $\pi_\vartheta$:

$$K \mid \pi_\vartheta = \phi_1(U), \quad S \mid \pi_\vartheta = \phi_2(U), \tag{15}$$

where $\phi_1$ and $\phi_2$ are arbitrary causal functions, acting as an extension to the path coefficient in (Pearl, 2009). These functions measure the intensity of the causal relationship from input to output. Then we have the impact from $K$ to $S$:

$$S \mid \pi_\vartheta = (\phi_2 \circ \phi_1^{-1})(K), \tag{16}$$

where $\phi_1^{-1}$ is the inverse function of $\phi_1$, whose intensity has a negative correlation with the intensity of $\phi_1$, due to the unidirectional deduction in causal inference. $(\phi_2 \circ \phi_1^{-1})$ shows the non-negligible correlation between $K$ and $S$. Consequently, in Fig. 1a, there is the frontdoor path $\pi_a = \{S \to Y\}$, so even if we exclude $S$ as a factor in logistic regression, the correlation $(\phi_2 \circ \phi_1^{-1})$ will cause impact from $S$ to $Y$:

$$Y \mid \pi_a = \phi_3(S). \tag{17}$$

where $(\phi_2 \circ \phi_1^{-1})$ is included in $S$, so this equation expressed when inferring from $K$ is:

$$Y \mid \pi_\vartheta \times \pi_a = (\phi_3 \circ \phi_2 \circ \phi_1^{-1})(K), \tag{18}$$

where $\phi_3$ merely depends on the distribution of $S$, which is fixed. However, the causal graph in Fig. 1b bypasses this frontdoor path by creating $\pi_b = \{S \leftarrow S' \to Y\}$. Similar as $\pi_a$, we have in $\pi_b$:

$$S \mid \pi_b = \phi_4(S'), \quad Y \mid \pi_b = \phi_5(S'), \tag{19}$$

$$Y \mid \pi_b = (\phi_5 \circ \phi_4^{-1})(S), \tag{20}$$

$$Y \mid \pi_\vartheta \times \pi_b = (\phi_5 \circ \phi_4^{-1} \circ \phi_2 \circ \phi_1^{-1})(K), \tag{21}$$

where $(\phi_5 \circ \phi_4^{-1})$ shows the correlation between $S$ and $Y$. Notably, $(\phi_5 \circ \phi_4^{-1})$ can be controlled by the auxiliary node $S'$, because $\phi_4$ and $\phi_5$ both take $S'$ rather than $S$ as inputs. This framework provides flexibility for users to balance fairness and accuracy by controlling whether $S'$ should more likely result in $S$ or $Y$. Specifically, if we strengthen the intensity of $\phi_4$, $S'$ then have a stronger causal effect to $S$. According to Eq. 20 and 21, both the correlation from $S$ to $Y$ and from $K$ to $Y$ are minimized, where the former contributes to counterfactual fairness, with a tradeoff of accuracy. Intuitively, we can consider minimizing correlations as controlling two information flows: one flowing from $S$ to $Y$ and the other flowing from $K$ to $Y$. Minimizing the correlations weakens the information flows, thus promoting counterfactual fairness and decreasing performance. So, we can conclude that introducing $S'$ improves counterfactual fairness and is naturally made to control information flows. Since we face the challenge of realizing this control process, we develop a control node $S''$ to tackle it.

### 3.3 $S''$: THE CONTROL NODE

Now, we introduce $S''$, where we design a custom loss that connects $S'$ and $S''$, acting as the key factor supporting information flow control. It minimizes the distance between $S'$ and $S''$:

$$\mathcal{L}_c(S', S'') = \frac{1}{D} \sum_{i=1}^{D} \|S_i' - S_i''\|_2^2, \tag{22}$$

where $D$ is the training dataset length, and $\|\cdot\|_2$ is the Euclidean norm (L2 norm). Here, we aim to deduce $S''$ as the descendant of $Y$ for calculating $\mathcal{L}_c(S', S'')$. In Pearl's SCM theory, deduced variables often represent hidden factors that cannot be directly observed. Estimating the posterior distribution of these latent variables is challenging (Kingma, 2013; Blei et al., 2017). Therefore, the implementation of the causal graph resorts to the ELBO technique (Jordan et al., 1999), which defines a guide model equipped with an assumed posterior distribution to fit the rules of the causal graph. In our case, we realize the Evidence Lower Bound (ELBO) loss as:

$$\mathcal{L}_{\text{ELBO}} = -\log p(\mathbf{O}) + \text{KL}(q(\mathbf{Z}|\mathbf{O}) \,\|\, p(\mathbf{Z}|\mathbf{O})), \tag{23}$$

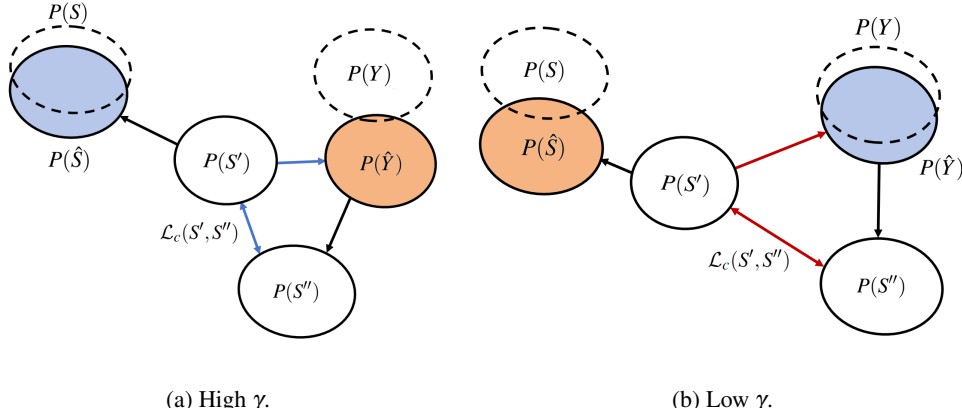

(a) High $\gamma$.          (b) Low $\gamma$.

Figure 2: The distribution mapping regarding $\mathcal{L}_c(S', S'')$, which can be seen as a probability inference perspective of the partial causal graph in Fig. 1b, where blue arrows mean the distribution parity are tightened, red arrows mean loosened, the dashed line circle is the true distribution, the full line circle is the inferred distribution. Blue circles are close to true distributions, and orange circles are far from true distributions. Note that the $\gamma$ is positively related to the effect of $\mathcal{L}_c(S', S'')$, so the constraint of $\mathcal{L}_c(S', S'')$ in Fig 2a is tighter, in Fig 2b is looser.

where $\mathbf{Z} = \{K, S', S''\}$ are the latent variables, $p(\mathbf{O})$ is prior, $q(\mathbf{Z}|\mathbf{O})$ is the approximate posterior distribution that tends to fit the true posterior distribution $p(\mathbf{Z}|\mathbf{O})$, $\text{KL}[\cdot \parallel \cdot]$ is the Kullback-Leibler (KL) divergence. The formula expresses that $\mathcal{L}_{\text{ELBO}}$ is the negative marginal log-likelihood $\log p(\mathbf{O})$ and the KL divergence between $q(\mathbf{Z}|\mathbf{O})$ and $p(\mathbf{Z}|\mathbf{O})$. Minimizing $\mathcal{L}_{\text{ELBO}}$ effectively minimizes the KL divergence, which in turn provides a better approximation of the true posterior $p(\mathbf{Z}|\mathbf{O})$.

Here the overall loss is defined as: $\mathcal{L} = \mathcal{L}_{\text{ELBO}} + \gamma \cdot \mathcal{R}(\mathcal{L}_c(S', S''))$, where $\gamma$ is the hyper-parameter that balances the two losses, $\mathcal{R}$ is a normalization scale ensuring $\mathcal{L}_{\text{ELBO}}$ and $\mathcal{L}_c(S', S'')$ are initialized with the same order of magnitude.

To understand why $S''$ can control the information flows, we need to rethink the ELBO technique from the implementation perspective. Specifically, what we do in the training process is to use ELBO to construct an approximate posterior distribution $q(\mathbf{Z}|\mathbf{O})$ that fits the true posterior distribution $p(\mathbf{Z}|\mathbf{O})$. After obtaining $q(\mathbf{Z}|\mathbf{O})$, during the inference, we can predict $\hat{Y}$ and $\hat{S}$ based on the approximate distribution. Note that $\hat{S}$ is different from observable variable $S$, where $\hat{S}$ is inferred from $q(S'|\mathbf{O})$.

Next, to distinguish the behavior of probability inference from causal inference, we notate the distributions of the nodes by $P(\cdot)$, and the distributions will simplify $(\cdot \mid \mathbf{o})$ expression. Fig. 2 shows the distribution mapping under different $\gamma$. Different from causal inference, probability inference is a method that focuses on correlation rather than causal relations. For example, although $S$ is a descendent of $S'$ in the causal graph, $P(S)$ will impact the inference result of $P(S')$ because the correlation between $S$ and $S'$ is mutual.

In the scenario of probability inference, to guarantee counterfactual fairness, we should ensure the causal relationship between $S'$ and $S$ is $S' \to S$ according to Fig. 1b. Applying it to probability inference, when we infer from $P(S')$ to $P(\hat{S})$, $P(\hat{S})$ should approximate $P(S)$. Therefore, $\text{KL}(P(S) \parallel P(\hat{S}))$ is a valuable property to estimate counterfactual fairness. For accuracy, we can estimate it as $\text{KL}(P(Y) \parallel P(\hat{Y}))$.

The effect of $\mathcal{L}_c(S', S'')$ in distribution manner is minimizing $\text{KL}(P(S'') \parallel P(S'))$. Now we can view the loss from two distinct perspectives:

- Fairness: Since both $S$ and $Y$ are descendants of $S'$, it is challenging for $S'$ to simultaneously infer both $\hat{S}$ and $\hat{Y}$ that closely match their true distributions. The decreased accuracy in fitting $P(Y)$ with $P(\hat{Y})$ creates an opportunity for $S'$ to better infer $P(\hat{S})$, thereby minimizing

$\text{KL}(P(\hat{S}) \parallel P(S))$. Consequently, this reduction in KL divergence enhances counterfactual fairness.

- Accuracy: since we formulate a deduction from $Y$ to $S''$ in causal graph (Pearl, 1995), the posterior distribution of $S''$ are constrained by $\hat{Y}$, i.e., $\text{KL}(P(\hat{Y}) \parallel P(S'')) < l$, where $l$ is positive. Because of the triangle inequality for KL divergence, we have:

$$\text{KL}(P(\hat{Y}) \parallel P(S')) \leq l + \text{KL}(P(S'') \parallel P(S')), \tag{24}$$

which demonstrates the upper bound of $\text{KL}(P(\hat{Y}) \parallel P(S'))$ is constrained by $\text{KL}(P(S'') \parallel P(S'))$, therefore minimizing $\text{KL}(P(S'') \parallel P(S'))$ also expect to minimize $\text{KL}(P(\hat{Y}) \parallel P(S'))$. This minimization will lead to $P(\hat{Y})$ aligning more closely with $P(S')$, which may compromise its alignment with $P(Y)$, resulting in a trade-off in accuracy.

**Connection between KL divergence and causal functions:** We observe that the greater the intensity of causal functions, the fewer distribution parities between inferred and true variables. For example, more intense $\phi_4$ with less $\text{KL}(P(\hat{S}) \parallel P(S))$, and more intense $\phi_5$ with less $\text{KL}(P(\hat{Y}) \parallel P(Y))$. Therefore, minimizing $\mathcal{L}_c(S', S'')$ can be viewed as a maximized causal intensity in $\phi_4$ and minimized causal intensity in $\phi_5$. According to Eq. 20 and 21, within path $\pi_{\vartheta} \times \pi_b$, the causal intensity of $(\phi_5 \circ \phi_4^{-1})$ is minimized. This minimization indicates less correspondence between $S$ and $Y$, contributing to counterfactual fairness, and less correspondence between $K$ and $Y$, compromising predicted accuracy.

**The benefit of $S''$ compared to $\hat{Y}$:** The purpose of using $S''$ in the custom loss rather than $\hat{Y}$ is to provide additional flexibility in controlling the influence of $S'$ on both $Y$ and $S$. By minimizing the distance between $S'$ and $S''$, the model can dynamically adjust the extent to which $S'$ influences both $Y$ and $S$ during the optimization process. Compared to directly minimizing the distance between $S'$ and $\hat{Y}$, this approach allows $S'$ to influence the prediction more subtly through the intermediate node $S''$. This gives the model greater freedom to prioritize fairness while maintaining performance.

## 4 EXPERIMENTS

### 4.1 EXPERIMENT SETTINGS

**Baselines:** To investigate the effectiveness of our framework in learning counterfactually fair predictors, we compare the proposed framework with multiple state-of-the-art methods. First, we briefly introduce all the compared baseline methods and their settings:

- **Constant Predictor:** It produces constant output. We obtain it by finding a constant minimizing the mean squared error (MSE) loss on the training data.
- **Full Predictor:** It takes $\mathbf{X}$ and $S$ as input for prediction.
- **Unaware Predictor:** It takes $\mathbf{X}$ as input for prediction to achieve fairness through unawareness (Dwork et al., 2012).
- **Counterfactual Fairness Predictors:** Fair-K (Kusner et al., 2017) reaches counterfactual fairness using the latent variables and non-descendants of the sensitive attribute in the prediction model. CLAIRE (Ma et al., 2023) involves creating a counterfactually fair dataset through augmentation and using a carefully designed loss function to ensure fairness during model training.

For baselines Full, Unaware, and Counterfactual Fairness Predictors, we use linear regression for regression and logistic regression for classification. Details about implementations, including datasets, environments, and hyper-parameters, are in Appendix B.

**Evaluation Metrics:** Generally speaking, the evaluation metrics consider two different aspects: prediction performance and counterfactual fairness. To measure the model prediction performance, we employ the widely used metrics - Root Mean Square Error (RMSE) (Chai et al., 2014) and Mean Absolute Error (MAE) (Yuan, 2022) for regression tasks and accuracy for classification tasks. To evaluate different methods concerning counterfactual fairness, we compare the distribution divergence of the predictions made on different counterfactuals in synthetic or real-world datasets. Detailed information about how to generate these counterfactuals is in the Appendix C. If a predictor is

counterfactually fair, the distributions of the predictions under different groundtruth counterfactuals are expected to be the same. Here, we use two distribution distance metrics (including Wasserstein-1 distance (Wass) (Chen et al., 2017) and Maximum Mean Discrepancy (MMD) (Long et al., 2015; Shalit et al., 2017)) to measure the distribution divergence. We compute the divergence of prediction distributions in every pair of counterfactuals ($S \leftarrow s$ and $S \leftarrow s^*$, $\forall s' \neq s$), then take the average value as the final result. The smaller the average values of MMD and Wass are, the better a predictor performs in counterfactual fairness.

## 4.2 BASELINE STUDY

**Baselines on synthetic datasets:** For a better measurement of counterfactual fairness, we generate a synthetic dataset for each real-world dataset, where we will *italicize* the synthetic datasets. Details about generating these synthetic datasets are in Appendix C. The results are shown in Table 1, where RMSE and MAE are performance metrics, and MMD and Wass are fairness metrics. Compared with Constant, Full, and Unaware baselines that omit the definition of counterfactual fairness, we showcase considerably better fairness. Compared with Fair-K and CLAIRE, we demonstrate not only better performance but also surpassing fairness.

Table 1: The comparison on synthetic datasets among Constant, Full, Unaware, Fair-K (Kusner et al., 2017), CLAIRE (Ma et al., 2023) and EXOC (Ours) on Law school (Krueger et al., 2021) and Adult (Becker & Kohavi, 1996) dataset.

| Method | *Law school* | | | | *Adult* | | |
| --- | --- | --- | --- | --- | --- | --- | --- |
| | RMSE ($\downarrow$) | MAE ($\downarrow$) | MMD ($\downarrow$) | Wass ($\downarrow$) | Accuracy ($\uparrow$) | MMD ($\downarrow$) | Wass ($\downarrow$) |
| Constant | $0.938_{\pm 0.004}$ | $0.759_{\pm 0.006}$ | $0.000_{\pm 0.000}$ | $0.000_{\pm 0.000}$ | $0.737_{\pm 0.006}$ | $0.000_{\pm 0.000}$ | $0.000_{\pm 0.000}$ |
| Full | $0.862_{\pm 0.005}$ | $0.689_{\pm 0.005}$ | $278.918_{\pm 25.814}$ | $69.248_{\pm 6.136}$ | $0.807_{\pm 0.005}$ | $52.515_{\pm 3.757}$ | $6.116_{\pm 0.637}$ |
| Unaware | $0.900_{\pm 0.008}$ | $0.726_{\pm 0.007}$ | $40.256_{\pm 3.187}$ | $10.256_{\pm 1.187}$ | $0.804_{\pm 0.008}$ | $19.732_{\pm 2.480}$ | $2.004_{\pm 0.478}$ |
| Fair-K | $0.894_{\pm 0.006}$ | $0.718_{\pm 0.006}$ | $4.313_{\pm 0.393}$ | $3.733_{\pm 0.267}$ | $0.745_{\pm 0.002}$ | $3.597_{\pm 0.256}$ | $1.553_{\pm 0.173}$ |
| CLAIRE | $0.897_{\pm 0.002}$ | $0.719_{\pm 0.002}$ | $6.717_{\pm 0.492}$ | $4.073_{\pm 0.139}$ | $0.748_{\pm 0.005}$ | $4.760_{\pm 0.275}$ | $1.584_{\pm 0.203}$ |
| EXOC | $0.874_{\pm 0.003}$ | $0.702_{\pm 0.003}$ | $3.824_{\pm 0.553}$ | $3.590_{\pm 0.259}$ | $0.760_{\pm 0.005}$ | $2.958_{\pm 0.124}$ | $1.428_{\pm 0.095}$ |

**Baselines on real-world datasets:** the result is shown in Table 2. Although compared with synthetic dataset results, we observe that the majority of the baselines demonstrate degradation in fairness and accuracy, we are still able to surpass the counterfactual-aware models in both performance and fairness and are fairer than counterfactual-unaware models. This observation demonstrates the robustness of our method in real-world scenarios.

Table 2: The comparison on real-world datasets among Constant, Full, Unaware, Fair-K (Krueger et al., 2021), CLAIRE (Ma et al., 2023) and EXOC (Ours) on Law school (Krueger et al., 2021) and Adult (Becker & Kohavi, 1996) dataset.

| Method | Law school | | | | Adult | | |
| --- | --- | --- | --- | --- | --- | --- | --- |
| | RMSE ($\downarrow$) | MAE ($\downarrow$) | MMD ($\downarrow$) | Wass ($\downarrow$) | Accuracy ($\uparrow$) | MMD ($\downarrow$) | Wass ($\downarrow$) |
| Constant | $0.940_{\pm 0.005}$ | $0.762_{\pm 0.004}$ | $0.000_{\pm 0.000}$ | $0.000_{\pm 0.000}$ | $0.724_{\pm 0.007}$ | $0.000_{\pm 0.000}$ | $0.000_{\pm 0.000}$ |
| Full | $0.883_{\pm 0.004}$ | $0.701_{\pm 0.005}$ | $574.013_{\pm 104.789}$ | $82.746_{\pm 8.298}$ | $0.791_{\pm 0.007}$ | $78.392_{\pm 5.723}$ | $6.989_{\pm 0.738}$ |
| Unaware | $0.917_{\pm 0.005}$ | $0.731_{\pm 0.007}$ | $48.738_{\pm 3.891}$ | $12.384_{\pm 1.542}$ | $0.800_{\pm 0.009}$ | $21.729_{\pm 2.573}$ | $2.425_{\pm 0.492}$ |
| Fair-K | $0.904_{\pm 0.005}$ | $0.723_{\pm 0.005}$ | $5.341_{\pm 0.412}$ | $3.980_{\pm 0.275}$ | $0.727_{\pm 0.004}$ | $4.381_{\pm 0.214}$ | $1.619_{\pm 0.175}$ |
| CLAIRE | $0.910_{\pm 0.003}$ | $0.735_{\pm 0.003}$ | $7.891_{\pm 0.502}$ | $4.095_{\pm 0.146}$ | $0.737_{\pm 0.004}$ | $5.140_{\pm 0.309}$ | $1.671_{\pm 0.224}$ |
| EXOC | $0.902_{\pm 0.005}$ | $0.720_{\pm 0.004}$ | $4.739_{\pm 0.553}$ | $3.879_{\pm 0.236}$ | $0.748_{\pm 0.004}$ | $3.891_{\pm 0.095}$ | $1.575_{\pm 0.098}$ |

## 4.3 ABLATION STUDY

### 4.3.1 ABLATION ON $\gamma$

We perform an ablation study on $\gamma$, shown in Tab. 3. This experiment evaluates the effect of controlling fairness-accuracy balance, running on synthetic datasets. The results show that as $\gamma$ increases from 1 to 2, we can observe the performance gradually decreases, but the counterfactual

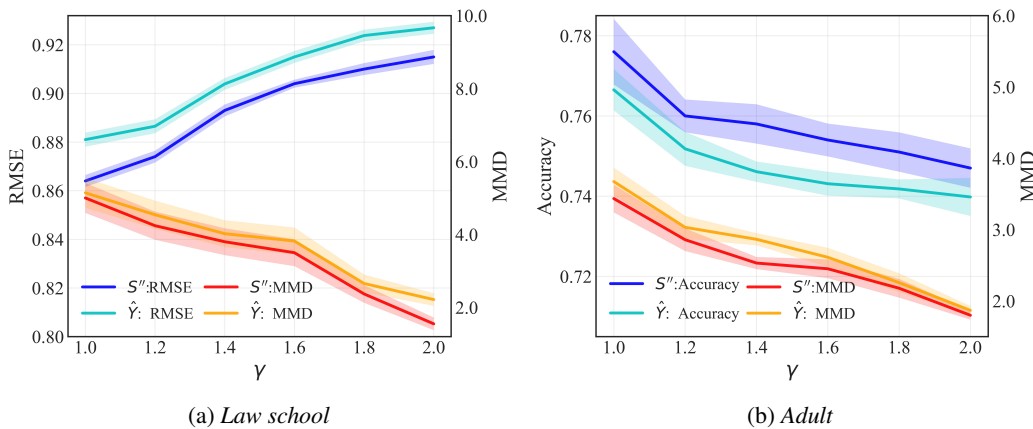

(a) *Law school*    (b) *Adult*

Figure 3: The ablation study on $S''$ and $\hat{Y}$

fairness gradually increases. Also, we observe a better fairness-accuracy tradeoff, i.e., an increased fairness without sacrificing much accuracy. We attribute this to the introduction of the auxiliary node $S'$, which serves as intrinsic information capable of deducing $S$. The result aligns with our theoretical analysis in Section 3.3, where $S''$ node and the custom loss $\mathcal{L}_c(S', S'')$ can control the fairness-accuracy tradeoff. We observe that when $\gamma = 1.2$, there is generally an excellent balance between accuracy and fairness. Therefore, we set $\gamma = 1.2$ in our experiments.

Table 3: The ablation study on $\gamma$.

| $\gamma$ | Law school | | | | Adult | | |
|---|---|---|---|---|---|---|---|
| | RMSE ($\downarrow$) | MAE ($\downarrow$) | MMD ($\downarrow$) | Wass ($\downarrow$) | Accuracy ($\uparrow$) | MMD($\downarrow$) | Wass ($\downarrow$) |
| 1 | $0.875_{\pm 0.006}$ | $0.698_{\pm 0.006}$ | $4.489_{\pm 0.571}$ | $3.905_{\pm 0.305}$ | $0.765_{\pm 0.006}$ | $3.532_{\pm 0.283}$ | $1.539_{\pm 0.245}$ |
| 1.2 | $0.867_{\pm 0.003}$ | $0.706_{\pm 0.003}$ | $3.832_{\pm 0.623}$ | $3.580_{\pm 0.256}$ | $0.760_{\pm 0.005}$ | $2.961_{\pm 0.124}$ | $1.426_{\pm 0.095}$ |
| 1.4 | $0.886_{\pm 0.003}$ | $0.724_{\pm 0.005}$ | $3.377_{\pm 0.452}$ | $3.352_{\pm 0.253}$ | $0.755_{\pm 0.006}$ | $2.628_{\pm 0.107}$ | $1.352_{\pm 0.089}$ |
| 1.6 | $0.900_{\pm 0.002}$ | $0.728_{\pm 0.005}$ | $3.089_{\pm 0.421}$ | $3.203_{\pm 0.251}$ | $0.757_{\pm 0.005}$ | $2.458_{\pm 0.109}$ | $1.297_{\pm 0.098}$ |
| 1.8 | $0.903_{\pm 0.003}$ | $0.731_{\pm 0.005}$ | $2.034_{\pm 0.322}$ | $2.890_{\pm 0.241}$ | $0.751_{\pm 0.006}$ | $2.068_{\pm 0.085}$ | $1.204_{\pm 0.089}$ |
| 2 | $0.909_{\pm 0.003}$ | $0.735_{\pm 0.004}$ | $1.342_{\pm 0.121}$ | $2.824_{\pm 0.204}$ | $0.746_{\pm 0.006}$ | $1.792_{\pm 0.074}$ | $1.184_{\pm 0.069}$ |

### 4.3.2 ABLATION ON $S''$ AND $\hat{Y}$

We perform an ablation study on whether $S''$ or $\hat{Y}$ should be used in the custom loss, i.e., $\mathcal{L}_c(S', S'')$ or $\mathcal{L}_c(S', \hat{Y})$, where the experiment runs on synthetic datasets and the results are shown in Fig. 3. The results show that when we apply $S''$ in the custom loss, we can observe an around 0.02 RMSE decrease on the Law school dataset and an around 0.02 Accuracy increase on the Adult dataset, indicating a better performance. This observation aligns with our analysis in Section 3.3. Moreover, the fairness metrics are slightly better when we apply $S''$. Therefore, we find it necessary to use $S''$ in the custom loss $\mathcal{L}_c(S', S'')$.

## 5 CONCLUSION

This paper introduces EXOC, a novel framework aimed at achieving counterfactual fairness while addressing the limitations of existing approaches. The key innovation lies in the revelation of intrinsic properties that are overlooked in previous works, through the introduction of auxiliary node $S'$ and control node $S''$. We demonstrate that they increase counterfactual fairness and also provide more refined control over the flow of intrinsic information beneath the concept of fairness and accuracy. Moreover, detailed analysis and extensive experimental evaluations on both synthetic and

real-world datasets demonstrate the framework's effectiveness, showing that EXOC outperforms state-of-the-art models in improving counterfactual fairness without sacrificing much accuracy.

Future work could explore scaling the framework to more complex datasets and simplifying its implementation for broader use. Theoretically, the connections between causal inference and its probability implementations are also of great interest. Developing more efficient optimization techniques for balancing the trade-off between fairness and accuracy, especially in high-dimensional data, could improve scalability and performance.

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
