# A   GENERAL DESIGN

Our framework can be adapted to various scenarios, particularly for deep or multi-layer causal models. The extended scenario is shown in Fig. 4. Generally, $S'$ serves as a mediator, replacing the direct causal relationship between the sensitive attribute $S$ and the target variable $Y$. Moreover, multi-layer causal relationships demonstrate that our framework can extend to complex problem settings such as computer vision (Krizhevsky et al., 2017) and natural language processing (Vaswani et al., 2017), which potentially requires multiple steps for generating the final answer.

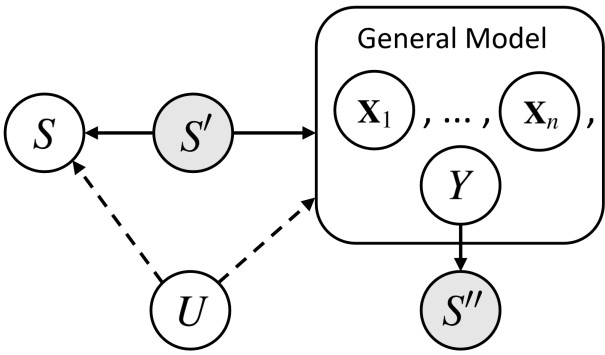

Figure 4: The generate causal model framework, where $\mathbf{X}_1$ to $\mathbf{X}_n$ denotes the layer of the causal relationship from $S$ to the related variables. For example, the Law school dataset is a special case of the framework, where $\mathbf{X}_1$ contains $\mathbf{X}$; $\mathbf{X}_2$ contains $K$ in Fig. 1.

# B   IMPLEMENTATION DETAILS

## B.1   DATASETS

**Law School (Krueger et al., 2021).** This dataset includes academic information from students at 163 law schools. We aim to predict each student's first-year average grade (FYA), making this a regression task. Race is treated as the sensitive attribute, while grade-point average (GPA) and entrance exam scores (LSAT) are the two observed features. This study focuses on individuals identified as White, Black, or Asian. The dataset comprises 20,412 instances.

**Adult (Becker & Kohavi, 1996).** The UCI Adult Income dataset contains census data for various adults, and the goal is to predict whether their income exceeds $50K$ per year. Race is considered the sensitive attribute $S$, and income is the prediction label $Y$, making this a binary classification task. We focus on individuals identified as White, Black, or Asian-Pac-Islander. In addition to race being the sensitive attribute, five other attributes are used for prediction. The dataset consists of 31,979 instances.

## B.2   HYPER-PARAMETERS AND ENVIRONMENTS

**Hyperparameter Settings:** For the two datasets, we split the training, validation, and test set as 80%, 10%, and 10%. All the presented results are on the test data. We set the number of training epochs as 8000, $\gamma = 1.2$; All the experiments have five independent runs.

**Environments:** The models are trained offline using PyTorch (Paszke et al., 2019) and executed on a machine equipped with an AMD EPYC 7763 64-Core Processor CPU @ 4.00GHz and an NVIDIA RTX 6000 Ada Generation GPU, running the Ubuntu 22.04.3 LTS operating system. The experiments run on the Conda environment and Docker container. We will release our Conda environment and Docker container upon publication. We attach our code in the Supplementary Materials.

## C  SYNTHETIC DATASET GENERATION

The synthetic dataset generation module is based on a Variational Auto-Encoder (VAE) (Kingma, 2013) with an encoder-decoder structure. Specifically, the encoder in the VAE takes $\{\mathbf{X}, Y\}$ as input, encodes them into a latent embedding space, and then the decoder reconstructs the original data $\{\mathbf{X}, Y\}$ with the embeddings $H$ and sensitive attribute $S$. ($H$ is the output of the VAE bottleneck layer to generate counterfactuals) Note that $S$ is only used as an input of the decoder to enable counterfactual generation in later steps. The reconstruction loss $\mathcal{L}_r$ is:

$$\mathcal{L}_r = \mathbb{E}_{q(H|\mathbf{X},Y)} \left[ -\log\left(p(\mathbf{X}, Y \mid H, S)\right) \right] + \text{KL}\left[ q(H \mid \mathbf{X}, Y) \parallel p(H) \right] \tag{25}$$

where $p(H)$ is a prior distribution, e.g., standard normal distribution $\mathcal{N}(0, I)$, $q(H \mid \mathbf{X}, Y)$ is the posterior approximation distribution. To eliminate the causal effect of $S$ on $H$, we introduce the Distribution Matching (Ma et al., 2023) technique by minimizing the dependency between them. In particular, we minimize the Maximum Mean Discrepancy (MMD) (Long et al., 2015; Shalit et al., 2017) among the embedding distributions of different sensitive subgroups. The loss function of training the counterfactual dataset generation model with distribution matching is as follows:

$$\min \mathcal{L}_r + \tau \frac{1}{N_p} \sum_{s \neq \tilde{s}} \text{MMD}(P(H \mid s), P(H \mid \tilde{s})) \tag{26}$$

where $N_p = \frac{|S| \times (|S| - 1)}{2}$ is the number of pairs of different sensitive attribute values, and $|S|$ is the number of different sensitive attribute values. The second term is the distribution matching penalty, which aims to achieve $P(H \mid S = s) = P(H \mid S = \tilde{s})$ for all pairs of different sensitive subgroups $(s, \tilde{s})$. Here, $\tau \geq 0$ is a hyperparameter that controls the importance of the distribution balancing term. Consequently, we can create a synthetic dataset for each real-world dataset to test the counterfactual scenarios better.