# OpenReview forum: "Towards counterfactual fairness through auxiliary variables"
_ICLR.cc/2025/Conference — ICLR 2025 Poster_

### Official Review · Reviewer_ZZ7B · 2024-10-27

**Soundness:** 4
**Presentation:** 3
**Contribution:** 3
**Rating:** 8
**Confidence:** 4

**Summary:**

The paper presents a novel framework called EXOC designed to achieve counterfactual fairness using auxiliary variables. By introducing an auxiliary node and a control node, the framework uncovers information from intrinsic properties that previous works have overlooked. EXOC demonstrates improved counterfactual fairness and offers finer control over the flow of intrinsic information, enhancing both fairness and accuracy. The framework is evaluated on synthetic and real-world datasets, showing superior performance compared to state-of-the-art models in terms of both counterfactual fairness and accuracy.

**Strengths:**

1. The theoretical analysis is compelling. The authors first outline an ideal model scenario, clearly demonstrating improved counterfactual fairness. They then extend their analysis beyond these assumptions to broader contexts, showing consistent performance improvements. The connection to information flow is creative, highlighting how the presence of $U$￼ leads to information leakage from $K$￼to $S$￼, while introducing $S'$￼ helps regulate the flow from ￼$S$ to $Y$￼.

2. The visual representations in Figure 2 effectively illustrate the impact of the control node. The clear differentiation between ￼$P(\hat{S})$ and $P(S)$￼ offers a fresh perspective on fairness issues, making the framework’s approach to counterfactual fairness more intuitive.

3. The paper is well-written, with clear and coherent presentation of research concepts. The experiments are thorough and well-designed, providing strong evidence to support the effectiveness of the proposed framework.

**Weaknesses:**

The introduction could be more comprehensive by including additional related works. While the authors discuss counterfactual fairness within the ML domain, there is a growing body of research that could complement the current submission, such as:

[1] Garg, Sahaj, et al. “Counterfactual fairness in text classification through robustness.” Proceedings of the 2019 AAAI/ACM Conference on AI, Ethics, and Society, 2019.

[2] Madras, David, et al. “Fairness through causal awareness: Learning causal latent-variable models for biased data.” Proceedings of the Conference on Fairness, Accountability, and Transparency, 2019.

[3] Zuo, Zhiqun, Mahdi Khalili, and Xueru Zhang. “Counterfactually fair representation.” Advances in Neural Information Processing Systems 36, 2023.

Including these references would strengthen the discussion of related works.

In Section 3.2.1, the authors provide an example to illustrate improvements in counterfactual fairness, further extended in Section 3.2.2. However, some aspects of the explanation could be clearer—please see the questions below for more details.

It is not clear why the authors chose a probabilistic model to implement the causal model, given that deterministic models (e.g., MLPs with activation functions) are often effective for inference tasks.

Some experimental settings and hyper-parameters could be better reported to ensure clarity and reproducibility.

**Questions:**

1. The introduction of causal functions involves the presence of $U$, but its specific contribution to the analysis is unclear. I also have concerns about the path analysis in your framework, as it appears to condition on specific paths (e.g., $\pi_{\vartheta}$￼ and $\pi_{b}$). Could other unconsidered paths also affect counterfactual fairness? If so, could these alternative pathways introduce unintended changes to fairness outcomes? How does your framework address or mitigate these potential effects?

2. Deterministic models like MLP and ResNet-50 are widely used due to their simplicity and effectiveness in optimization. Could you clarify the rationale for opting for probabilistic models and probabilistic inference in this context? Additionally, please elaborate on the relationship between probabilistic inference and causal inference in your framework. How do these two forms of inference interact, and what benefits does this interaction offer for achieving fairness and accuracy?

3. The trade-off between fairness and accuracy is well-documented in fairness research, often implying that gains in accuracy come at the cost of fairness. Could you elaborate on how controlling information flow differs from managing this trade-off, and provide justification for the distinction?

---

> ### Author Response · Authors · 2024-11-24
> **Official Comment by Authors (Part 1)**
>
> Thank you for your suggestions in detail and your contribution to improving the clarity of our work. We would be delighted to address any questions and look forward to further discussions and feedback.
>
> W1. The introduction could be more comprehensive by including additional related works.
>
> Thanks for providing the following works, we incorporate them in our introductions, and showcase that although all three papers focus on counterfactual fairness, aiming to address how sensitive attributes (e.g., gender, race) influence model predictions, they lack the explicit consideration of which factor results in the sensitive attribute. Compared with their work, we explicitly focus on the reasoning of sensitive attributes to reach the counterfactual fairness, and to the best of our knowledge, this reasoning is not explored in previous works.
>
> W2. Some experimental settings and hyper-parameters could be better reported to ensure clarity and reproducibility.
>
> We write the implementation details, including datasets (Law school [4] and Adult [2]), hyper-parameters and environments in the Appendix B to ensure clarity and reproducibility. Moreover, we will open-source our implementations and environments upon publication.
>
> Q1. Concerns about the path analysis in your framework, as it appears to condition on specific paths. Could other unconsidered paths also affect counterfactual fairness?
>
> In Sections 3.2.2, we describe the process of selecting critical paths including $\pi_{\vartheta}$, $\pi_a$ and $\pi_{b}$. We focus on these paths because they are most relevant to the fairness-accuracy trade-off, where the effect transmitting from $S$ to $Y$ affects fairness most. Other paths either do not pass through $S$ or have more indirect effect from $S$ to $Y$, which are not the primary contributing paths to fairness issues. Therefore, we don’t focus on them in our theoretical analysis.
>
> Q2. Could you clarify the rationale for opting for probabilistic models and probabilistic inference in this context? Additionally, please elaborate on the relationship between probabilistic inference and causal inference in your framework. How do these two forms of inference interact, and what benefits does this interaction offer for achieving fairness and accuracy?
>
> We provide a distinction between probability inference and causal inference from line 308-312, where probability inference is a method that focuses on correlation rather than causal relations. There are two major distinction between probability inference and causal inference:
>
> 1. Probability inference allows bidirectional reasoning, but causal inference is inherently unidirectional.
>
> 2. Causal inference introduces the concept of intervention, which is explicitly modeled and accounted for. In contrast, probability inference does not sufficiently consider the effects of interventions.
>
> In conclusion, causal inference can be described as a logic-thinking machine, where you can deduce output from the input, just like probability inference. However, according to Pearl’s theory [4], it further considers the layer of intervention, emphasizing that intervening on the output should not alter the distribution of the input. To make the connection between Pearl’s theory to machine learning framework more explicit [5], our implementation utilizes probability inference and incorporates ELBO technique to realize counterfactual fairness [3, 5, 6].

---

> ### Author Response · Authors · 2024-11-24
> **Official Comment by Authors (Part 2)**
>
> Q3. The difference between the trade-off between fairness and accuracy and controlling the information flow
>
> The relationship between controlling the information flow and the trade-off between fairness and accuracy lies in the challenge of theoretical difference these two objectives. In traditional approaches, it is difficult to theoretically demonstrate the trade-off between fairness and accuracy, and adjusting hyper-parameters alone often fails to achieve either high fairness or high accuracy effectively. Our method, as shown in Sections 3.2.2 and 3.3, provides a theoretical framework to control the trade-off between fairness and accuracy, allowing for flexibility in different application scenarios. For example, in accuracy-priority problems, such as medical diagnosis or object detection, a lower $\gamma$ value can be used to maintain accuracy. Conversely, in fairness-priority problems, such as racially oriented job applications or graduate school admissions, a higher $\gamma$ value can enhance fairness among different demographic groups.
>
> ## Reference
> [1] Barry Becker and Ronny Kohavi. Adult. UCI Machine Learning Repository, 1996. DOI: https://doi.org/10.24432/C5XW20.
>
> [2] Jing Ma, Ruocheng Guo, Aidong Zhang, and Jundong Li. Learning for counterfactual fairness
> from observational data. In Proceedings of the 29th ACM SIGKDD Conference on Knowledge
> Discovery and Data Mining, 2023.
>
> [3] Vincent Grari, Sylvain Lamprier, and Marcin Detyniecki. Adversarial learning for counterfactual fairness. Machine Learning, 2023.
>
> [4] Pearl, Judea. "Causal inference in statistics: An overview.", 2009.
>
> [5] Kusner, Matt J., et al. "Counterfactual fairness." Advances in neural information processing systems, 2017.
>
> [6] Hyemi Kim, Seungjae Shin, JoonHo Jang, Kyungwoo Song, Weonyoung Joo, Wanmo Kang, and Il.Chul Moon. Counterfactual fairness with disentangled causal effect variational autoencoder. In Association for the Advancement of Artificial Intelligence, 2021.

---

> ### Comment · Reviewer_ZZ7B · 2024-11-25
>
> Thanks for the detailed reply. The most concerns are addressed. I'd love to increase my score to 8.

---

> > ### Author Response · Authors · 2024-11-25
> > **Official Comment by Authors**
> >
> > Thank you for your insightful comments regarding the probability inference, path analysis, and information flow aspects of our work. We have carefully addressed these points in our responses and greatly appreciate the opportunity to refine this critical part of our paper further. We are grateful for your observations, and we are confident that they have significantly improved the clarity of our work, making it both more comprehensible and impactful for the broader research community.

---

### Official Review · Reviewer_HYEE · 2024-10-31

**Soundness:** 1
**Presentation:** 1
**Contribution:** 1
**Rating:** 3
**Confidence:** 4

**Summary:**

This paper aims to achieve counterfactual fairness through the introduction of auxiliary variables $S’$ and control node $S’’$.

**Strengths:**

The introduction of auxiliary variables $S’$ and control node $S’’$ is interesting.

**Weaknesses:**

- The authors assert in the introduction part that their framework can "enhance fairness without compromising accuracy." However, the evidence supporting this claim is not readily discernible in the paper. A clearer demonstration or justification of this statement is needed.

- The methodology section lacks clarity in several areas. For instance, in lines 199-201, the statement "As $\alpha(s*−s)$ is the counterfactual parity that plays a more important role than the standard deviation, we showcase that δa>δb" is not intuitively clear. Additionally, there is no distinction between probability inference and causal inference, leaving the reader confused about the framework. Furthermore, the rationale behind choosing $S''$ over $\hat{Y}$ is unclear. Lines 345-350 do not provide enough context or reasoning.

- The experimental results raise several questions. Specifically, why does the EXOC model outperform the Full model in accuracy when $\gamma=1$ on the Law School dataset? This outcome appears coincidental, and further analysis or explanation is necessary.

- The paper does not include comparisons with other models aimed at achieving counterfactual fairness, such as mCEVAE (Pfohl et al., 2019), DCEVAE (Kim et al., 2021), ADVAE (Grari et al., 2023), or CFGAN (Xu et al., 2019). A comparison with these established methods is necessary for evaluating the effectiveness and innovation of the proposed framework.

Stephen R. Pfohl, Tony Duan, Daisy Yi Ding, and Nigam H Shah. Counterfactual reasoning for fair clinical risk prediction. In Machine Learning for Healthcare Conference, 2019.

Hyemi Kim, Seungjae Shin, JoonHo Jang, Kyungwoo Song, Weonyoung Joo, Wanmo Kang, and Il.Chul Moon. Counterfactual fairness with disentangled causal effect variational autoencoder. In AAAI, 2021.

Vincent Grari, Sylvain Lamprier, and Marcin Detyniecki. Adversarial learning for counterfactual fairness. Machine Learning, 2023.

Depeng Xu, Yongkai Wu, Shuhan Yuan, Lu Zhang, and Xintao Wu. Achieving causal fairness through generative adversarial networks. In IJCAI, 2019.

**Questions:**

Please refer to the weaknesses above.

---

> ### Author Response · Authors · 2024-11-24
> **Official Comment by Authors (Part 1)**
>
> Thank you for your constructive feedback and valuable suggestions to enhance the clarity of our work. Below, we address the concerns raised and provide clarifications where misunderstandings may have arisen, particularly regarding Section 3.2.1 (in W2 (1)) and the introduction of probability inference in Section 3.3 (in W2 (2)). We are committed to further clarifying any remaining ambiguities in the revised manuscript.
>
> W1. The evidence of the assertion in the introduction that the framework can "enhance fairness without compromising accuracy."
>
> We appreciate the opportunity to clarify our statement. By “without compromising accuracy,” we mean that our framework achieves enhanced fairness while maintaining a competitive level of accuracy. Our evidence comprises both experimental and theoretical analyses. Experimentally, compared to the baselines that focus on counterfactual fairness, such as Fair-K and CLAIRE, we showcase that can achieve a better accuracy; and compared to full model, we don’t sacrifice much accuracy. Theoretically, in Section 3.2.2, we provide a clear demonstration of the control of fairness-accuracy balance (control here means that we have the formalization to achieve a only-fair model, a only-accurate model or any model in between). If we combine the observations and analysis together, we can further observe that the existence of $S’$ actually helps increasing the fairness-accuracy balance while allowing controlling, which explains why we enhance fairness without sacrificing much accuracy.
>
> W2 (1). In lines 199-201, the statement "As $\alpha(s^*-s)$ is the counterfactual parity that plays a more important role than the standard deviation, we showcase that $\delta_a > \delta_b$ is not intuitively clear.
>
> We thereby provide a clarification of why $\alpha(s^*-s)$ is the counterfactual parity that plays a more important role than the standard deviation.
>
> First, we have clarified in section 3.2.2 that the causal functions are an extension of path coefficient demonstrated in 3.2.1, where adding the Gaussian noise to the random variables can be viewed as a special case of the causal intensity in 3.2.2. For example, if the Gaussian noise of $K$ in $Y = \alpha S + \beta K$ is quite large, then the causal intensity from $K$ to $Y$ is weak. In this manner, considering the case that the causality is well-modeled in the SCM and the exhibited, the causal intensity is large enough, so the Gaussian noise is quite small compared to the direct interference on $S$. We will further clarify it in our revised version.
>
> W2 (2). There is no distinction between probability inference and causal inference, leaving the reader confused about the framework.
>
> We provide a distinction between probability inference and causal inference from line 308-312, where probability inference is a method that focuses on correlation rather than causal relations. There are two major distinction between probability inference and causal inference:
>
> 1. Probability inference allows bidirectional reasoning, but causal inference is inherently unidirectional.
>
> 2. Causal inference introduces the concept of intervention, which is explicitly modeled and accounted for. In contrast, probability inference does not sufficiently consider the effects of interventions.
>
> In conclusion, causal inference can be described as a logic-thinking machine, where you can deduce output from the input, just like probability inference. However, according to Pearl’s theory [4], it further considers the layer of intervention, emphasizing that intervening on the output should not alter the distribution of the input. To make the connection between Pearl’s theory to machine learning framework more explicit [5], our implementation utilizes probability inference and incorporates ELBO technique to realize counterfactual fairness [3, 5, 6].
>
> W2 (3). The rationale behind choosing $S’’$ over $\hat{Y}$
>
> The task of $S'$ is to infer the distributions related to $S$ and $Y$. However, since both $S$ and $Y$ are descendants of $S'$, directly inferring $S'$ encounters conflicts. By introducing $S''$ and optimizing $L_c(S', S'')$, $S'$ is better able to decouple its relationship with $S$ and $Y$, namely, dynamically adjust the extent to which $S’$ influences both $Y$ and $S$, to help $S’$ better optimize to both variables.
>
> W3. The experimental issue on the $\gamma$ ablation study
>
> After running five more rounds of the experiment, we observe in most cases of $\gamma=1$, our method is below but approximating the Full model, and the fairness-accuracy balance monotony is evident, supporting our theoretical analysis of controlling the tradeoff between fairness and accuracy, and an enhanced balance compared with baseline models.
>
> (continue in the next comment)

---

> > ### Comment · Reviewer_HYEE · 2024-11-27
> >
> > Thank you to the authors for their response. However, I find the responses to W2 and W4 unconvincing.
> >
> > Regarding the exemplar clarity issue raised in W2, the authors again attempt to provide an intuitive explanation, but a more rigorous explanation is necessary. In Section 3, there is no rigorous mathematical derivation connecting the observed correlations to the desired causal effects. This lack of formalization makes the methodology section (or the so-called “theoretical part”) unclear. Furthermore, it remains unclear how a counterfactual inference problem is reduced to a probability inference problem. Counterfactuals are inherently non-identifiable, so how can you ensure that the proposed method achieves counterfactual fairness rather than causal fairness at the intervention level? These unresolved issues make the proposed method unconvincing to me.
> >
> > Similarly, the experimental concerns raised in W4 remain unaddressed. For a newly proposed VAE-based model aimed at achieving counterfactual fairness, it is crucial to compare its performance with other similar VAE-based approaches addressing counterfactual fairness. In addition to experimental performance, differences in design, particularly the objective function (which typically includes a shared ELBO term), should be thoroughly discussed. Without such comparisons and discussions, it is difficult to evaluate the validity of the method.
> >
> > Given these unresolved concerns, I will maintain my score.

---

> ### Author Response · Authors · 2024-11-24
> **Official Comment by Authors (Part 2)**
>
> W4. Other baselines aimed at achieving counterfactual fairness
>
> Thank you for your suggestions on baselines. We have provided the baselines aiming at achieving counterfactual fairness, including Fair-K [1] and CLAIRE [2]; and baselines on constant / full / fairness with unawareness classifiers in Section 4.2. Here is the additional evalutions with Adversarial Learning for Counterfactual Fairness (ALCF) [3] based on the Law school dataset:
>
> |  &nbsp;**Method**   |      &nbsp;&nbsp;&nbsp;&nbsp;&nbsp;&nbsp; **RMSE ↓**   |       &nbsp; &nbsp;&nbsp;&nbsp;&nbsp;&nbsp;**MAE ↓**   |           &nbsp;&nbsp;&nbsp;&nbsp;&nbsp;&nbsp;&nbsp;&nbsp;&nbsp;&nbsp;**MMD ↓**         |        &nbsp;&nbsp;&nbsp;&nbsp;&nbsp;&nbsp;&nbsp;&nbsp;**Wass ↓**        |
> |:--------------:|:--------------:|:--------------:|:--------------:|:--------------:|
> | **Constant** | 0.938 ± 0.004 | 0.759 ± 0.006 | 0.000 ± 0.000    | 0.000 ± 0.000     |
> | **Full**     | 0.862 ± 0.005 | 0.689 ± 0.005 | 278.918 ± 25.814 | 69.248 ± 6.136    |
> | **Unaware**  | 0.900 ± 0.008 | 0.726 ± 0.007 | 40.256 ± 3.187   | 10.256 ± 1.187    |
> | **Fair-K**   | 0.894 ± 0.006 | 0.718 ± 0.006 | 4.313 ± 0.393    | 3.733 ± 0.267     |
> | **ALCF**     | 0.884 ± 0.006 | 0.721 ± 0.006 | 7.681 ± 0.924    | 5.974 ± 0.286     |
> | **CLAIRE**   | 0.897 ± 0.002 | 0.719 ± 0.002 | 6.717 ± 0.492    | 4.073 ± 0.139     |
> | **EXOC**     | 0.874 ± 0.003 | 0.702 ± 0.003 | 3.824 ± 0.553    | 3.590 ± 0.259     |
>
> Since ALCF focuses on leveraging adversarial training to generate counterfactual fairness data (note that we have utilized VAE to generate counterfactual fairness data, details are in Appendix C ), its limitations lie in the lack of emphasis on model design and the insufficient consideration of latent variables that influence the sensitive attribute to further enhance counterfactual fairness.
>
> Thank you for providing the additional materials. The majority of existing work focus on VAE and counterfactual-fair data generation. We give further clearance on the following works:
>
> 1. Stephen R. Pfohl, Tony Duan, Daisy Yi Ding, and Nigam H Shah. Counterfactual reasoning for fair clinical risk prediction. In Machine Learning for Healthcare Conference, 2019.
>
> This paper leverages a VAE-based framework for generating counterfactual samples, primarily targeting fairness in high-dimensional healthcare datasets. Because their code is closed-source and focuses on medical datasets, we did not compare them directly. Theoretically, similar to ALCF, they primarily use VAE, which is utilized to generate counterfactual-fair datasets, agnostic to latent variables, and the formulation of causal models.
>
> 2. Hyemi Kim, Seungjae Shin, JoonHo Jang, Kyungwoo Song, Weonyoung Joo, Wanmo Kang, and Il.Chul Moon. Counterfactual fairness with disentangled causal effect variational autoencoder. In Association for the Advancement of Artificial Intelligence, 2021.
>
> The paper Counterfactual Fairness with Disentangled Causal Effect Variational Autoencoder (DCEVAE) focuses on achieving counterfactual fairness by disentangling exogenous uncertainty into latent variables. Because their code is closed-source, we did not compare them directly. Theoretically, they propose a DCEVAE to generate counterfactual fairness datasets, which will be useful for downstream tasks. Our method not only considers the counterfactual fair dataset generation but also explores the fundamental causes of sensitive attributes within causal models.
>
> 3. Depeng Xu, Yongkai Wu, Shuhan Yuan, Lu Zhang, and Xintao Wu. Achieving causal fairness through generative adversarial networks. In International Joint Conference on Artificial Intelligence, 2019.
>
> In this paper, the authors process binary data preprocessing and binarize each attribute to reduce the complexity of causal graph discovery, so their training framework can not explicitly compare with ours. Theoretically, our framework leverages the ELBO technique and efficiently trains the final classifier, while their approach prioritizes generating high-quality fair data using two generators and two discriminators in an adversarial training setup, which is more time-intensive.
>
> ## Reference
> [1] Matt J Kusner et al. "Counterfactual fairness." Advances in neural information processing systems, 2017.
>
> [2] Jing Ma et al. "Learning for counterfactual fairness
> from observational data." In Proceedings of the 29th ACM SIGKDD Conference on Knowledge
> Discovery and Data Mining, 2023.
>
> [3] Vincent Grari, Sylvain Lamprier, and Marcin Detyniecki. "Adversarial learning for counterfactual fairness." Machine Learning, 2023.
>
> [4] Pearl, Judea. "Causal inference in statistics: An overview." 2009.
>
> [5] Kusner, Matt J., et al. "Counterfactual fairness." Advances in neural information processing systems, 2017.
>
> [6] Hyemi Kim et al. "Counterfactual fairness with disentangled causal effect variational autoencoder." In Association for the Advancement of Artificial Intelligence, 2021.

---

> ### Author Response · Authors · 2024-12-02
> **Official Comment by Authors (Part 3)**
>
> Thank you for your thoughtful feedback and follow-up questions. We appreciate the opportunity to clarify these questions. Here is our detailed response:
>
> 1. In Section 3, there is no rigorous mathematical derivation connecting the observed correlations to the desired causal effects. This lack of formalization makes the methodology section (or the so-called “theoretical part”) unclear. Furthermore, it remains unclear how a counterfactual inference problem is reduced to a probability inference problem.
>
> In Section 3, we present our framework in the order of linear cases,  general cases, and implementations, where Section 3.2.2 introduces arbitrary causal functions to bridge the gap between counterfactual fairness and probability inference implementations.
>
> Inspired by your suggestions, we offer a more detailed explanation of the relationship between observed correlations and the desired causal effects. First, let us revisit our claim in Section 3.2.2, where we argue that the arbitrary causal functions, from a causal inference perspective, support the advantage of $S'$ in balancing fairness and performance. Section 3.2.2 aims primarily to provide an “explanation” or “understanding” of how data are generated or an ontological perspective of the problem [1] (Is classifying on the non-descendant variable of $S$ intrinsically exclude the effect from $S$ ? The answer is No.) However, the probability inference is epistemic since it assumes an observer that can find the correlation between the input and output, regardless of the ontology itself. However, although the starting points from the two perspectives are inherently different, we provide an important connection between them, which we will formalize.
>
> **Theorem**. Consider the input $X$, output $Y,$ and the causal function $\phi$ from $X$ to $Y$, then we assume a good classifier $f$ that can capture the correlation between causal input $X$ and output $Y$, yielding $\hat{Y}$. For an ideal causal model, the infinite approximation of causal intensity $I(\phi) \rightarrow +\infty $ is a sufficient but not necessary condition of predicted correlation error $\text{KL}(P(Y)\parallel P(\hat{Y})) \rightarrow 0$.
>
> **Proof. (Sufficiency)** $I(\phi) \rightarrow +\infty $ means the causal relationship from $X$ to $Y$ is intense, whose correlation is easy to capture in ideal causal models. Since the classifier $f$ can accurately capture the correlation between causal input $X$ and output $Y$, which is evidently $\text{KL}(P(Y)\parallel P(\hat{Y})) \rightarrow 0$.
>
> **(Necessity)** $\text{KL}(P(Y)\parallel P(\hat{Y})) \rightarrow 0$ only means that $f$ is a accurate classifier from $X$ to $Y$, but not indicating specific causal relationships. For example, we can consider a mediator $A$ that can simultaneously result in $X$ and $Y$, producing the perfect classifier.
>
> This theorem can be interpreted from different perspectives. Another perspective is that the smaller $I(\phi)$ is, the larger $\text{KL}(P(Y)\parallel P(\hat{Y}))$ will be, indicating the information deduced from $X$, $\hat{Y}$ is less dependent to $Y$.
>
> This theorem sheds light on the theoretical grounding of Section 3.3, i.e., **using probability inference to handle our construction of the causal model.** Here are some details to clarify: why don’t we directly use some mechanisms based on causal inference to handle it? The answer is that, as we have clarified, causal models are logic-thinking machines instead of real-world implementations. For example, imagine you are a creator that creates a certain causal rule, but changing into the observer perspective, we only have data that have already been created, so we can not tell what the creators actually do (but we can discover the causal relationships, which is called the causal discovery [1], that’s beyond the scope of this paper).

---

> ### Author Response · Authors · 2024-12-02
> **Official Comment by Authors (Part 4)**
>
> 2. Counterfactuals are inherently non-identifiable, so how can you ensure that the proposed method achieves counterfactual fairness rather than causal fairness at the intervention level? These unresolved issues make the proposed method unconvincing to me.
>
> The distinction between **counterfactual fairness** and **causal fairness at the intervention level** ranges from formula to implementation. In our framework, the counterfactual fairness is defined as follows:
>
> $P\left( Y_{S \leftarrow s } = y\mid \mathbf{o} \right) = P\left( Y_{S \leftarrow s'} = y\mid \mathbf{o} \right), \quad \forall s' \neq s,$
>
> however, causal fairness at the intervention level is defined as follows:
>
> $P(Y\mid do(S=s)) = P(Y\mid do(S=s’)), \quad \forall s' \neq s.$
>
> The causal effect measures the change of setting one value for a variable compared with a reference. In contrast, the counterfactual effect measures what if a variable $S$ whose value **is $s$** is set to $s’$. Therefore, causal fairness measures **the impact** of a sensitive attribute on the decision; counterfactual fairness answers the question: what is a value is changed **from s** (factual world) **to s’** (counterfactual world) [4].
>
> For the identification of the counterfactuals, a mathematical way to determine is the **ID*** **algorithm** [2], as it determines and provides methods to transform counterfactual effects into computable probabilistic expressions. If the effect can not be strictly computed, then it is unidentifiable. In this case, as explicitly stated in [3], we can measure **bounds for counterfactual effects or path-specific effects** even in unidentifiable situations.
>
> 3. Similarly, the experimental concerns raised in W4 remain unaddressed. For a newly proposed VAE-based model aimed at achieving counterfactual fairness, it is crucial to compare its performance with other similar VAE-based approaches addressing counterfactual fairness. In addition to experimental performance, differences in design, particularly the objective function (which typically includes a shared ELBO term), should be thoroughly discussed. Without such comparisons and discussions, it is difficult to evaluate the validity of the method.
>
> Thank you for raising the question; **we respectfully disagree that our method is VAE-based**. VAE is only used for counterfactual data generation to better measure the fairness and performance of the model. In fact, we develop a novel causal model achieving counterfactual fairness. Therefore, our baselines should be previous works addressing counterfactual fairness, as we have provided those baselines, including Fair-K [4], CLAIRE [5] and ALCF [6], and so on. **Also, we highlight it is not necessary that approaches with VAE include a shared ELBO term.** ELBO is a specific probability inference technique, while VAE is used for data generation. For example, in our real-world experiments, we use the ELBO technique, but we use real-world data rather than VAE-generated data.
>
> As shown in Section 1, our work challenges the assumption in previous studies that sensitive attributes should not be causally influenced by any other variables, and develop a novel causal model.  In Section 3, we present our framework from special linear cases, general cases to the implementations.  With your valuable review, we conducted additional experimental comparisons and theoretical discussions, which allowed us to explore the concerns more thoroughly. We truly appreciate your thoughtful comments and look forward to your feedback.
>
> ## Reference
> [1] Pearl, Judea. "Models, reasoning and inference." Cambridge, UK: CambridgeUniversityPress 19.2 (2000): 3.
>
> [2] Shpitser, Ilya, and Judea Pearl. "Complete identification methods for the causal hierarchy." (2008).
>
> [3] Wu, Yongkai, et al. "Pc-fairness: A unified framework for measuring causality-based fairness." Advances in neural information processing systems 32 (2019).
>
> [4] Matt J Kusner et al. "Counterfactual fairness." Advances in neural information processing systems, 2017.
>
> [5] Jing Ma et al. "Learning for counterfactual fairness from observational data." In Proceedings of the 29th ACM SIGKDD Conference on Knowledge Discovery and Data Mining, 2023.
>
> [6] Vincent Grari, Sylvain Lamprier, and Marcin Detyniecki. "Adversarial learning for counterfactual fairness." Machine Learning, 2023.

---

### Official Review · Reviewer_LPsA · 2024-11-03

**Soundness:** 2
**Presentation:** 2
**Contribution:** 3
**Rating:** 5
**Confidence:** 2

**Summary:**

This paper introduces the EXOC framework, a causal model leveraging auxiliary and control nodes to enhance counterfactual fairness while maintaining predictive accuracy. It shows improved fairness compared to existing methods, validated on both synthetic and real-world datasets.

**Strengths:**

1) EXOC introduces a new approach by utilizing auxiliary variables to capture latent information, improving counterfactual fairness and accuracy, which is new to my knowledge
2) Demonstrates competitive accuracy and fairness on benchmark datasets, outperforming prior models. Though in many cases the improvement is only with accuracy or fairness, not as impressive as claim in the intro.

**Weaknesses:**

1) The paper does not adequately explain the process for tuning hyperparameters, particularly within the causal structure. Specifically, the role of hyperparameter 𝛾 in balancing fairness and accuracy is crucial, yet the approach to fine-tuning this balance remains unclear. And I am wondering if a efficient tuning strategy exist as it involves a tradeoff.
2) The causal diagram in Figure 1, illustrating the roles of auxiliary nodes S' and control nodes S'' , lacks practical grounding. The authors could strengthen their framework by providing realistic examples or settings in which these nodes causally impact the system in a way that aligns with the proposed structure. This would improve interpretability and offer a clearer justification for the chosen causal relationships.
3) Apart from above two, a major weakness I would point out is the experiment section:

Although synthetic data is a valid approach, as we could assume a groundtruth, Generating synthetic data with a Variational Auto-Encoder (VAE) introduces additional assumptions and potential challenges, such as ensuring the VAE accurately represents counterfactual distributions. The authors do not report VAE performance metrics or provide convincing examples of generated counterfactuals, which makes it difficult to verify if the synthetic data truly captures the desired counterfactual fairness properties. Without this validation, the reliability of the synthetic evaluation is unclear.

Both the use of real-world and synthetic datasets is very limited. For example, other commonly used fairness benchmarks include insurance dataset, Crime dataset. More datasets would strengthen the validation and help demonstrate EXOC’s assumption robustness across diverse real-world settings.

**Questions:**

1) Could you provide more details on the approach to tuning hyperparameters, especially γ, within the causal structure? How sensitive are your fairness and accuracy results to changes in this parameter, and what guidelines would you recommend for tuning it on different datasets?

2) Could you offer a practical, real-world scenario where the auxiliary node S′ and control node S′′ would interact causally in the manner illustrated in Figure 1? Examples or applications with concrete causal relationships would help clarify the intended structure and interpretability of the model.

3) Why did you choose a VAE for synthetic data generation over more conventional methods in fairness literature? Additionally, could you provide performance metrics or examples demonstrating the validity of the VAE-generated counterfactuals?

---

> ### Author Response · Authors · 2024-11-24
> **Official Comment by Authors (Part 1)**
>
> Thank you for your suggestions in detail and your contribution to improving the clarity of our work. We would be delighted to address any questions and look forward to further discussions and feedback.
>
> W1/Q1. Methods for fine-tuning $\gamma$, and whether there is an efficient tuning strategy exists as it involves a trade-off:
>
> Yes, recent works have demonstrated that the trade-off between fairness and accuracy is monotonic in convex settings [2]. We also observe monotonic behavior in the theoretical analysis and experiments. The existence of these monotonic relationships makes adjusting the trade-off between fairness and accuracy significantly more convenient. Next, we will discuss how to regulate this trade-off in different scenarios.
>
> Specifically, $\gamma$ serves as an important hyper-parameter that controls the fairness-accuracy trade-off, and we explain why $\gamma$ has the capacity of this controlling in section 3.3. For specific methods for fine-tuning $\gamma$, we believe the trade-off will be different in different problem settings. There are two kinds of problems, accuracy-priority problems and fairness-priority problems, highlighting different task objectives. For example, in accuracy-priority problems such as medical diagnosis and object detection, $\gamma$ should be lower to maintain accuracy; in fairness-priority problems such as racial oriented job applications or graduate school admissions, $\gamma$ should be higher to enhance fairness among different demographic groups. So a appropriate value of $\gamma$ is personalized for different individual tasks. As for tasks in our experiment, where we mainly focus on fairness, we show that $\gamma$ around 1.2 performs well. For accuracy-priority problems, we expect $\gamma$ to be lower than 1.2.
>
> W2/Q2. The practical grounding and interpretability of auxiliary nodes S' and control nodes S'':
>
> Thank you for this constructive suggestion. We provide the practical grounding of $S’$ and $S”$. For auxiliary node $S’$, it serves as the mediator that simultaneously impacts $S$, $\mathbf{X}$ and $Y$. For example, in the Law School dataset, $S$ is the gender, $Y$ is the student’s first-year average grade, and $\mathbf{X}$ are some contributing factors such as GPA. In this case, $S’$ can be viewed as the student's domain knowledge. Given the domain knowledge of the student, we can show gender is only a conclusion of the domain knowledge rather than a determinant of the student’s first-year average grade. Similarly, $S”$ can be viewed as feedback for the student’s first-year average grade based on the first-year average grade expectations. In this manner, the model’s interpretability is also shown because if someone asks why the student’s first-year average score is high, we will first consider the domain knowledge of the student and view gender as a marginal conclusion. In the meantime, we can deduce the student's feedback based on the score.
>
> W3 (1)/Q3. The reason to choose a VAE for synthetic data generation over more conventional methods in fairness literature; Missing VAE performance metrics or convincing examples of generated counterfactuals, which makes it difficult to verify if the synthetic data truly captures the desired counterfactual fairness properties.
>
> As tailored in various works [3, 4, 5], VAE has been verified as an effective tool for generating samples that can better satisfy counterfactual fairness. We perform a statistical study on VAE to demonstrate the effectiveness of the VAE model.
>
> | &nbsp; **Epoch** |   &nbsp;&nbsp;&nbsp;&nbsp;**Sum**   |  &nbsp;&nbsp;&nbsp;&nbsp;&nbsp;&nbsp;**$\mathcal{L}_r$**  |     &nbsp;&nbsp;&nbsp;&nbsp;&nbsp;**_MMD_**     |
> |:---------:|:------------:|:------------:|:--------------------:|
> | **0**     | 126.7778    | 126.7774     |     4.0116e-4           |
> | **1000**  | 34.0371      | 34.0371      |   1.0037e-5         |
> | **2000**  | 14.1027      | 14.1027      |   7.9544e-9         |
> | **3000**  | 3.7271       | 3.7271       |   7.9366e-13         |
> | **4000**  | 2.9541       | 2.9541       |   5.9844e-14         |
> | **5000**  | 2.9272       | 2.9272       |   1.8285e-14         |

---

> ### Author Response · Authors · 2024-11-24
> **Official Comment by Authors (Part 2)**
>
> Details about the losses can be found in Appendix C, where the Sum is the sum of two loss terms. $\mathcal{L}_r$ ensures that the VAE decoder can regenerate the input data $\mathbf{X}$ and $Y$ based on the latent variable $H$ and the sensitive attribute $S$. **_MMD_** measures the causal effect of the sensitive attribute on the latent variable $H$. Consequently, a decrease in both $\mathcal{L}_r$ and **_MMD_** demonstrates that the VAE is able to generate the dataset that ensures the real-world distribution is accurately modeled, and changes in $S$ do not affect the generated distribution much (matches the definition of counterfactual fairness). Here we are listing  the examples of generated counterfactuals:
>
> |     &nbsp;&nbsp;&nbsp;&nbsp;&nbsp;&nbsp;&nbsp;&nbsp;Sample     |      &nbsp;&nbsp;&nbsp;Race      |      &nbsp;&nbsp;&nbsp;&nbsp;&nbsp;&nbsp;&nbsp;&nbsp;&nbsp;&nbsp;&nbsp;&nbsp;$\mathbf{X}$        |      &nbsp;&nbsp; $Y$        |
> |:--------------:|:--------------:|:--------------:|:--------------:|
> | **Real-World** |   Hispanic  | [39.0, 3.1, 0.78]  |   0.73   |
> | **Counterfactual1** |  White  | [39.4, 3.3, 0.78]  |   0.74   |
> | **Counterfactual2** |  Black | [38.8, 3.0, 0.80] |   0.72     |
>
> Where the generated samples are like Counterfactual1 and Counterfactual2, their sensitive attribute changes, but the related feature $\mathbf{X}$ and outcome $Y$ are similar to the original distribution.
>
> W3 (2). To demonstrate the effectiveness of EXOC, we add an experiment to show the effectiveness of different datasets. Here we provide the evaluation of the crime dataset [1], compared with different baselines:
> |  &nbsp;&nbsp;**Method**   |      &nbsp;&nbsp;&nbsp;&nbsp;&nbsp;&nbsp; **RMSE ↓**   |       &nbsp; &nbsp;&nbsp;&nbsp;&nbsp;&nbsp;**MAE ↓**   |           &nbsp;&nbsp;&nbsp;&nbsp;&nbsp;&nbsp;&nbsp;&nbsp;&nbsp;&nbsp;**MMD ↓**         |        &nbsp;&nbsp;&nbsp;&nbsp;&nbsp;&nbsp;&nbsp;&nbsp;**Wass ↓**        |
> |:--------------:|:--------------:|:--------------:|:--------------:|:--------------:|
> | **Constant** | 0.245 ± 0.010 | 0.173 ± 0.006 | 0.000 ± 0.000    | 0.000 ± 0.000     |
> | **Full**     | 0.167 ± 0.009 | 0.116 ± 0.012 | 359.503 ± 64.815 | 50.647 ± 6.157    |
> | **Unaware**  | 0.208 ± 0.012 | 0.143 ± 0.007 | 33.046 ± 4.153   | 12.752 ± 1.472  |
> | **Fair-K**   | 0.204 ± 0.007 | 0.140 ± 0.005 | 5.719 ± 0.351    | 4.051 ± 0.210     |
> | **EXOC**     | 0.192 ± 0.006 | 0.130 ± 0.004 | 3.980 ± 0.515    | 3.498 ± 0.187     |
>
> The experiment results show that we effectively demonstrate a better fairness-accuracy tradeoff, which further showcases the effectiveness of our method in various datasets.
>
> Additional comments for Q1: What is the sensitivity of hyper-parameters?
> For the $\gamma$ sensitivity, we performed the ablation study shown in Table 3. We show that within a reasonable range, the parameter is not sensitive. Moreover, as we have demonstrated the control of fairness-accuracy, adjusting $\gamma$ is a valuable tool for this control.
>
> ## Reference
> [1] D. Dua and C. Graff, “UCI ml repository,” http://archive.ics.uci.edu/ml, 2017.
>
> [2] Wu, Yongkai, Lu Zhang, and Xintao Wu. "Fairness-aware classification: Criterion, convexity, and bounds." arXiv preprint arXiv:1809.04737, 2018.
>
> [3] Ma, Jing, et al. "Learning for counterfactual fairness from observational data." Proceedings of the 29th ACM SIGKDD Conference on Knowledge Discovery and Data Mining. 2023.
>
> [4] Stephen R. Pfohl, Tony Duan, Daisy Yi Ding, and Nigam H Shah. Counterfactual reasoning for fair clinical risk prediction. In Machine Learning for Healthcare Conference, 2019.
>
> [5] Vincent Grari, Sylvain Lamprier, and Marcin Detyniecki. Adversarial learning for counterfactual fairness. Machine Learning, 2023.

---

> > ### Comment · Reviewer_LPsA · 2024-11-25
> >
> > Thank you for the detailed response and the additional experiments. I appreciate that other reviewers share the concerns regarding experimental limitations and some have adjusted their scores following the authors' new efforts. However, I align with HYEE's perspective that more extensive experimentation is necessary to evaluate the generalizability of the proposed causal diagram beyond two simple synthetic datasets and one real-world Crime dataset.
> >
> > Given this perspective, I will maintain my score and defer to the AC's decision.

---

> ### Author Response · Authors · 2024-11-26
> **Official Comment by Authors**
>
> Thank you for your comments regarding the experimentation in our work and for your constructive suggestions on the extensive evaluations. We would like to highlight that the majority of causal-based fairness studies focus on a few synthetic datasets or simpler real-world tabular datasets [1, 2, 3, 4], as they are well-suited for illustrating the key theoretical contributions. Similarly, our choice of datasets ensures clarity in showcasing both the theoretical and practical value of our approach. We appreciate your thoughtful review and look forward to hearing any further insights or suggestions.
>
> ## Reference
>
> [1] Jing Ma et al. "Learning for counterfactual fairness from observational data." In Proceedings of the 29th ACM SIGKDD Conference on Knowledge Discovery and Data Mining, 2023.
>
> [2] Vincent Grari, Sylvain Lamprier, and Marcin Detyniecki. "Adversarial learning for counterfactual fairness." Machine Learning, 2023.
>
> [3] Depeng Xu, Yongkai Wu, Shuhan Yuan, Lu Zhang, and Xintao Wu. Achieving causal fairness through generative adversarial networks. In IJCAI, 2019.
>
> [4] Matt J Kusner, Joshua Loftus, Chris Russell, and Ricardo Silva. Counterfactual fairness. Advances in neural information processing systems, 30, 2017.

---

### Official Review · Reviewer_5RPc · 2024-11-03

**Soundness:** 3
**Presentation:** 3
**Contribution:** 3
**Rating:** 8
**Confidence:** 5

**Summary:**

This work aims to address the issue that previous methods ignored the intrinsic information in the exogenous variable U that does not lead to discrimination in predicting labels, leading to degraded performance. They propose a new causal graph with latent variables S' and S''.

**Strengths:**

- The proposed framework allows a trade-off between utility and fairness by adjusting the strength of the correlation between S' and S''.
- In linear cases, they provide bounds on the counterfactual fairness error to show the benefit of S'. They also provide an analysis with unknown functions in the causal graph.
- Strong empirical performance across different datasets and settings.
- Ablation study on \gamma and S'' supported the analysis in Section 3.

**Weaknesses:**

- The proposed method heavily rely on the assumed causal graph in Fig. 1 (b). I wonder how general this causal graph can be.
- The experiments are only conducted on two small tabular datasets.

**Questions:**

1. How is validation/model selection performed for the experiments?
2. Why do the authors only consider S' and S''? Is there any other latent variable that can be considered in that are not in the causal graph of Fig. 1 (b)?

---

> ### Author Response · Authors · 2024-11-24
> **Official Comment by Authors (Part 1)**
>
> Thank you for carefully checking our paper and thinking of the generalization of our framework. We would be delighted to address any questions and look forward to further discussions and feedback.
>
> W1. The generalization of the proposed framework:
>
> Thank you for the insightful suggestion! After carefully rethinking the framework shown in Figure 1, we write a new section about General Design and attach it to Appendix A. We showcase that our framework can be adapted to various scenarios, particularly for deep or multi-layer causal models. The extended scenario is shown in Fig. 4. Generally, $S'$ serves as a mediator, replacing the direct causal relationship between the sensitive attribute $S$ and the target variable $Y$. Moreover, multi-layer causal relationships demonstrate that our framework can extend to complex problem settings such as computer vision and natural language processing, which potentially requires multiple steps for generating the final answer.
>
> W2. The experiments are only conducted on two small tabular datasets.
>
> To demonstrate the effectiveness of EXOC, we add an experiment to show the effectiveness of different datasets. Here we provide the evaluation of the crime dataset [1], compared with different baselines:
>
> |  &nbsp;**Method**   |      &nbsp;&nbsp;&nbsp;&nbsp;&nbsp;&nbsp; **RMSE ↓**   |       &nbsp; &nbsp;&nbsp;&nbsp;&nbsp;&nbsp;**MAE ↓**   |           &nbsp;&nbsp;&nbsp;&nbsp;&nbsp;&nbsp;&nbsp;&nbsp;&nbsp;&nbsp;**MMD ↓**         |        &nbsp;&nbsp;&nbsp;&nbsp;&nbsp;&nbsp;&nbsp;&nbsp;**Wass ↓**        |
> |:--------------:|:--------------:|:--------------:|:--------------:|:--------------:|
> | **Constant** | 0.245 ± 0.010 | 0.173 ± 0.006 | 0.000 ± 0.000    | 0.000 ± 0.000     |
> | **Full**     | 0.167 ± 0.009 | 0.116 ± 0.012 | 359.503 ± 64.815 | 50.647 ± 6.157    |
> | **Unaware**  | 0.208 ± 0.012 | 0.143 ± 0.007 | 33.046 ± 4.153   | 12.752 ± 1.472  |
> | **Fair-K**   | 0.204 ± 0.007 | 0.140 ± 0.005 | 5.719 ± 0.351    | 4.051 ± 0.210     |
> | **EXOC**     | 0.192 ± 0.006 | 0.130 ± 0.004 | 3.980 ± 0.515    | 3.498 ± 0.187     |
>
> The experiment results show that we effectively demonstrate a better fairness-accuracy tradeoff, which further showcases the effectiveness of our method in various datasets.

---

> ### Author Response · Authors · 2024-11-24
> **Official Comment by Authors (Part 2)**
>
> Q1. How is validation/model selection performed for the experiments?
>
> The validation and model selection processes for the experiments in the paper are described in Appendix B.2: Hyper-parameters and Environments. Here is the relevant information summarized:
>
> 1. The datasets were split into 80% training, 10% validation, and 10% test sets.
>
> 2. All reported results are based on the test set.
>
> For model selections, we will take Law school dataset as an example. First, we construct the EXOC causal model as in Fig. 1 (b) and using ELBO technique to implement the code. Next, from the dataset perspective, we train a VAE model to better measure the counterfactual effect. We use the VAE model because the real-world dataset are sometimes agnostic to the counterfactual scenarios, for example, a real-world Hispanic person have a good First-Year Average (FYA), but there is few Black/ White person with similar relative variables have a the same FYA. Therefore, we use the VAE to create those counterfactual scenarios and expect we can classify them correctly. Last, we use the knowledge (the latent variable $K$ generated by the causal model) and perform the logistic regression on the dataset to perform the experiments.
>
> Q2. Why do the authors only consider S' and S''? Is there any other latent variable that can be considered in that are not in the causal graph of Fig. 1 (b)?
>
> Our motivation is that most existing works assume sensitive attributes should not be causally influenced by any other variable [2, 3, 4]. The direct causal path from $S$ to $Y$, as demonstrated in Fig. 1 (a), is the primary source of bias. Therefore, the design aims to introduce $S’$ that segregates the direct bias from $S$; $S’’$ is a further consideration that assists $S’$ to control the balance between fairness and accuracy. And yes, we realize that there are numerous latent variables that we can consider, for example, there are latent variables that impact both $S$ and $\mathbf{X}$; we think these latent variables are not as relevant to fairness as $S’$ is. But it would be more than interesting to propose a more general form of $S’$, such as in different modalities and dimensions.
>
> ## Reference
> [1] D. Dua and C. Graff, “UCI ml repository,” http://archive. ics.uci.edu/ml, 2017.
>
> [2] Matt J Kusner, Joshua Loftus, Chris Russell, and Ricardo Silva. Counterfactual fairness. Advances in neural information processing systems, 30, 2017.
>
> [3] Richard Berk, Hoda Heidari, Shahin Jabbari, Michael Kearns, and Aaron Roth. Fairness in criminal justice risk assessments: The state of the art. Sociological Methods & Research, 2021
>
> [4] Ma, Jing, et al. "Learning for counterfactual fairness from observational data." Proceedings of the 29th ACM SIGKDD Conference on Knowledge Discovery and Data Mining. 2023.

---

> > ### Comment · Reviewer_5RPc · 2024-11-24
> >
> > Thanks for the reply. I think it helps. I will raise the score by 1.

---

### Author Response · Authors · 2024-12-04

We thank all reviewers for their time and valuable feedback on our paper. Below, we provide a global overview of the updates made during the rebuttal process:

1. **Clarified the general design and scalability of the auxiliary node \(S'\)**

   We provide a detailed discussion on the general design of our approach in Appendix A and expand on the scalability considerations of \(S'\). These updates aim to enhance the clarity and generalizability of our proposed approach.

2. **Enhanced baselines, ablation studies, and expanded experiments on crime datasets**

   We include additional explanations for the ablation studies and refined the VAE module details, including effectiveness measurement. Additionally, we extended experiments to the crime dataset and benchmarked against the ALCF baseline, demonstrating our method’s superior fairness-performance trade-offs.

3. **Clarified different fairness definitions**

   We provide a clear distinction between counterfactual fairness and causal fairness at the intervention level, discussing that the ID* algorithm can transform counterfactual effects into computable probabilistic expressions, and bounds can be estimated even in non-identifiable scenarios.

4. **Clarified causal model contributions**

   We clarify that we have developed a new causal model to realize counterfactual fairness, while VAE models are only for better measuring fairness and performance. Moreover, we highlight approaches with VAE don't need to include a shared ELBO term.

5.  **Expanded discussion on trade-offs and controlling the information flow**

    We clarify the differences between fairness-accuracy trade-offs and controlling information flow, showcasing how our method balances these aspects while maintaining theoretical rigor.

---

### Meta-Review · Area_Chair_Tug8 · 2024-12-13

**Metareview:**

In this work, the authors introduce additional, latent variables to obtain counterfactual fairness. All reviewers appreciated the novelty of this framing.

Some concerns were raised in terms of comparisons with other baselines, as well as limited experiments on real-world datasets. While I feel that addressing these comments would make the paper stronger, I also believe that this paper is an interesting contribution to the field of counterfactual fairness and therefore recommend acceptance.

**Additional Comments On Reviewer Discussion:**

The authors engaged in the discussion and provided a thourough rebuttal as well as performed additional experiments. The paper also had 2 advocates during the reviewer discussion, which signals an interest from the field to see the method published. I also agree with the arguments made in terms of the limited availability of real-world datasets to study counterfactual fairness.

---

### Decision · Program_Chairs · 2025-01-22

Accept (Poster)